# The Clinical Utility of the NETest in Patients with Small Intestinal Neuroendocrine Neoplasms (Si-NENs): A “Real-Life” Study

**DOI:** 10.3390/cancers16142506

**Published:** 2024-07-10

**Authors:** Julian Gertner, Marina Tsoli, Aimee R. Hayes, Luke Furtado O’Mahony, Faidon-Marios Laskaratos, Thomas Glover, Priyesh Karia, Mohsin F. Butt, Oliver Eastwood, Dalvinder Mandair, Martyn Caplin, Christos Toumpanakis

**Affiliations:** 1University College London Hospital, London NW1 2BU, UK; 2Neuroendocrine Tumour Unit, ENETS Centre of Excellence, Royal Free Hospital, London NW3 2QG, UK; martso.mt@gmail.com (M.T.); aimee.hayes@nhs.net (A.R.H.); oliver.eastwood@gstt.nhs.uk (O.E.); dalvinder.mandair@nhs.net (D.M.); martyn.caplin@nhs.net (M.C.); c.toumpanakis@ucl.ac.uk (C.T.); 3Department of Oncology, University of Oxford, Oxford OX1 2JD, UK; luke.furtadoomahony1@nhs.net; 4St Mark’s Hospital, London HA1 3UJ, UK; flaskaratos@nhs.net (F.-M.L.); thomas.glover1@nhs.net (T.G.); 5St. Peter’s Hospital, Ashford TW15 3AA, UK; 6NIHR Nottingham Biomedical Research Centre, Nottingham NG7 2UH, UK; mohsin.butt@nhs.net

**Keywords:** small intestinal neuroendocrine neoplasms, biomarker, NETest, prognosis

## Abstract

**Simple Summary:**

This study examines the validity of the NETest, a blood biomarker, in patients with small intestinal neuroendocrine neoplasms (Si-NENs). Two groups were studied: those with metastatic Si-NENs (Group 1) and those without macroscopic disease, post-operatively (Group 2). In Group 1, results showed that NETest scores were associated with progression-free survival (PFS) and overall survival (OS) in metastatic Si-NENs. Factors including tumour growth rate (TGR), elevated chromogranin A (CgA) levels, and presence of lung metastases were also predictive of PFS or OS. In Group 2, NETest monitoring after surgery may have potential in helping to predict early local or metastatic disease recurrence. This study suggests potential benefits of using NETest alongside other predictors for Si-NEN management. Further research with longer follow-up periods is needed to confirm the utility of the NETest.

**Abstract:**

Current biomarkers do not adequately predict the behaviour of neuroendocrine neoplasms (NENs). This study assessed the NETest, a multianalyte blood biomarker, in patients with small intestinal NENs (Si-NENs). We studied two patient groups: Group 1: metastatic Si-NENs (*n* = 102) and Group 2: post-operatively disease-free according to 68Ga-DOTATATE PET (*n* = 16). NETest scores were ≤20% (normal), 21–40% (low), 41–79% (intermediate), or ≥80% (high). Overall survival (OS) and progression-free survival (PFS) were assessed using the Kaplan–Meier method. Univariate and multivariate analyses were performed using the Cox proportional hazards model. In Group 1, the median NETest score was 40% (IQR: 33.3–46.7%). The NETest value (HR: 1.032, 95% CI: 1.003–1.062, *p* = 0.033) and high-risk NETest category (HR: 10.5, 95% CI: 1.35–81.7, *p* = 0.025) were independent predictors of PFS, along with presence of lung metastases, CgA levels > 10 × ULN, and tumour growth rate (TGR). Independent predictors of OS were the NETest value (HR: 1.035, 95% CI: 1.005–1.066, *p* = 0.024) and high-risk NETest category (HR: 15.2, 95% CI: 1.52–151, *p* = 0.02), along with presence of lung metastases and CgA levels > 10 × ULN. In Group 2, ROC analysis identified an AUC of 0.909 (95% CI: 0.75–0.100) for prediction of local or metastatic recurrence. Blood NETest scores were associated with PFS and OS in patients with metastatic Si-NENs, along with TGR, CgA > 10 × ULN, and presence of lung metastases.

## 1. Introduction

Neuroendocrine neoplasms (NENs) are rare and heterogeneous neoplasms with an estimated incidence of 2–6/100,000 per year [1]. They arise mainly in the gastrointestinal tract, the pancreas, and the bronchopulmonary system [2]. Small intestinal NENs (Si-NENs) represent 26% of all NENs and often present late with distant metastases [3,4]. Patients with Si-NENs may undergo curative surgery, even with liver metastases, while palliative debulking may be suggested for symptom control [4]. Systemic treatment options for non-resectable metastatic disease include long-acting somatostatin analogues (SSAs), everolimus, and peptide receptor radionuclide therapy (PRRT) [5]. SSAs are largely well tolerated; some adverse effects include injection site pain, steatorrhoea, and more significantly cholelithiasis [6].

Follow-up should be life-long and includes regular radiological examination and biomarker evaluation. Anatomical imaging using the Response Evaluation Criteria in Solid Tumors (RECIST) criteria displays limitations in NENs. RECIST criteria assesses disease progression and therapeutic response based on changes in tumour size and number without considering tumour density. This can result in overestimation of stable disease and underestimation of treatment response. Functional imaging with ^68^Ga-DOTATATE positron emission tomography (^68^Ga-DOTATATE PET) also has limitations such as reduced spatial resolution (5 mm) and difficulty in detecting marginal change in tumour size [7,8,9].

Over the years, several circulating biomarkers have been developed for the early diagnosis and follow-up of patients with NENs [10]. However, the currently available biomarkers represent a monoanalyte assessment of protein or amines that cannot reflect accurately the biological behaviour of these heterogeneous neoplasms [11]. Chromogranin A (CgA) has been the most important and the most commonly used blood biomarker, but limitations in assay reproducibility, sensitivity, and specificity have been extensively documented [4,10]. There is a need for novel multianalyte biomarkers that can accurately reflect disease activity, therapeutic efficacy, and to improve the early detection of recurrent and progressive disease.

The NETest is a multi-analyte algorithmic biomarker that analyses 51 gene transcripts. Its uses have been shown in NEN diagnosis and early detection of post-operative recurrence, as well as predicting the response to treatment with SSAs and PRRT [12,13,14,15,16,17,18]. The NETest displays high sensitivity (98%) and specificity (97%), superior to monoanalyte biomarkers such as CgA [19,20]. The accuracy for detecting Si-NENs has been calculated as 93% [19]. A recent meta-analysis showed that the NETest had an approximately 95% diagnostic accuracy and was 84.5–85.5% accurate in differentiating stable disease from progressive disease [13]; it may also detect disease progression one year before imaging studies [21]. Moreover, it has been shown that the post-operative recurrence in NEN patients could be predicted with NETest-positive in >94% of cases vs. 11% with CgA [20].

The aim of this prospective study was to evaluate the validity of the NETest, in real-life, as a biomarker for the early identification of tumour progression and assessment of risk for post-operative local or metastatic recurrence in patients with Si-NENs.

## 2. Materials and Methods

### 2.1. Patients

In this single centre prospective study, patients with Si-NENs were recruited from March 2019 to March 2020, and followed up until March 2021. We studied two groups of patients: (a) Group 1: patients with metastatic Si-NENs (*n* = 102) and (b) Group 2: patients operated for Si-NENs that were post-operatively disease-free based on ^68^Ga-DOTATATE PET results (*n* = 16). All patients had well differentiated NENs of small bowel primary diagnosed at the NEN multidisciplinary team meeting on the basis of histopathological and radiological confirmation. The tumour grade was determined according to the Ki-67 proliferation index based on the 2019 WHO classification [22].

In Group 1, the NETest was collected at a random point during their disease course and follow-up was conducted according to European Neuroendocrine Tumor Society (ENETS) guidelines [5] for surveillance protocols with sequential radiological assessment using computed tomography (CT) and/or magnetic resonance imaging (MRI) and functional imaging with ^68^Ga-DOTATATE PET. Only patients with radiologically evident disease on CT and/or MRI scans were included. Radiological assessment was based on RECIST 1.1 criteria [8] and was performed at three different time points according to the NETest collection time point: pre-NETest (T−), refers to scans performed within 6–18 months prior to the NETest; peri-NETest (T0), refers to scans performed within 4 months of the NETest; and post-NETest (T+), refers to scans showing radiological progression post-NETest or the most recent scans in the case of no progression.

In Group 2, the NETest sample was collected in patients with a negative ^68^Ga-DOTATATE PET 3 months post-surgery for a Si-NEN. Follow-up was conducted with standard cross-sectional imaging with CT or MRI every 4–6 months and with a ^68^Ga-DOTATATE PET after the first year.

Imaging studies were evaluated by two independent radiologists who were blinded on clinical data. Treatment decisions were carried out independently of the NETest results. All patients provided informed consent. Ethics approval at UCL Royal Free Hospital Biobank was granted by Research Ethics Committee, reference 16/WA/0289.

### 2.2. NETest Measurement

Blood samples (10 mL) were collected in ethylenediaminetetraacetic acid (EDTA) tubes (BD Vacutainer Venous Blood Collection Tubes, BD Diagnostics, Franklin, NJ, USA). Aliquots of whole blood were stored at −80 °C within 2 h of collection (samples immediately stored on ice/4 °C after sampling) and were sent to Sarah Cannon Research Institute for processing and polymerase chain reaction (PCR). Wren laboratories performed analysis remotely via an application programming interface (API).

In a two-step protocol, mRNA was isolated from the whole blood samples and the PCR values of the 51 gene markers were analysed and converted to an activity score between 0 and 100. The upper limit of normal (ULN) is 20, scores of 21–40 are associated with low disease activity, 41–79 represent intermediate disease activity and ≥80 reflect high disease activity [23]. This score is then segregated into clinically relevant categories of normal, low, medium, and high risk. A score ranging from 21 to 40% reflects “stable” disease, whereas a score > 40% represents “progressive” disease.

### 2.3. Calculation of TGR

TGR was calculated and expressed as a percentage change in tumour volume over one month (%/m) using a previously published formula [24].
TGR = 100 × (exp[TG] − 1)TG = (3 × ln[D2/D1])/time (months)
where TG = tumour growth, D1 = tumour size at date 1, D2 = tumour size at date 2, and time (months) = (date 2 − date 1 + 1)/30.44.

Target lesions were measured at three different time points relative to the NETest: pre-NETest (T−), peri-NETest (T0), and post-NETest (T+). T0 refers to the time of the scan performed within 4 months of the NETest. T− refers to the time of the scan performed within 6–18 months prior to T0. T+ refers to the time of the scan showing radiological progression post-NETest or the most recent scan if remained free of progression. TGR was calculated at T0 by comparing imaging between T− and T0.

### 2.4. Statistics

All statistical analyses (frequencies, descriptive statistics, x^2^, Kaplan–Meier curves, log-rank tests, and Cox-regression analysis) were conducted with the StataMP 16 software package (Stata Corporation, College Station, TX, USA) and R software version 4.0.1 (R Foundation for Statistical Computing, Vienna, Austria).

Data were expressed as mean values ± standard error of the mean if normally distributed, otherwise they were displayed as median and interquartile range (IQR). Categorical variables were presented as percentages, and comparison was performed using Fisher’s exact test. Receiver Operator Characteristic (ROC) analysis was used to assess the diagnostic accuracy of the NETest. Overall survival (OS) and progression-free survival (PFS) were analysed using the Kaplan–Meier method. To calculate the PFS and OS, the date of NETest collection was used as the baseline. Cox regression analysis was used to assess the association between various clinical and histopathological variables, and the PFS or OS with both univariate (UVA) and multivariate (MVA) analysis. Correlation matrices and VIF were used to detect multicollinearity, and, if present, variables were analysed in separate models. For continuous variables, linearity was checked using deviance and martingale residuals. Proportional hazards assumptions were checked using Schoenfeld residuals. If a hazard ratio could not be calculated due to pseudoseparation or small sample size, Firth’s penalised log likelihood was used to obtain hazard ratios. Tests were two-sided, *p* < 0.05 was considered statistically significant, and the 95% confidence interval (CI) was provided for survival estimates.

## 3. Results

The study groups’ demographics and clinicopathological characteristics at baseline are shown in Table 1.

### 3.1. Group 1

#### 3.1.1. Patient and Tumour Characteristics

One hundred and two patients with metastatic Si-NENs were analysed in total (Table 1). The mean age was 65.3 ± 10.2 years, and the female-to-male ratio was 0.96 (50/52). All patients were diagnosed with well-differentiated neuroendocrine tumours. The WHO grade for Si-NENs was confirmed as grade 1 in 59 patients (58%) and grade 2 in 36 patients (35%) while Ki-67 was not available in 7 patients. The majority of patients (81%) had liver metastases while 13% had lung and 36% had bone metastases. Sixty-five (64%) patients had surgical resection of the primary tumour, ninety (88%) had treatment with SSAs, and thirty (29%) had prior PRRT. During follow-up, 86 (84%) patients were treated with SSAs, 14 (14%) received PRRT, and no patients had everolimus, sunitinib, or chemotherapy. A total of 22 patients (22%) had serum CgA levels increased above 10-times the ULN (>10 × ULN), while 55 (55%) patients had CgA levels <5 × ULN.

The median follow-up duration was 18 months (range: 11–23 months) after NETest collection. The median PFS and OS of this cohort was not reached during the follow-up time.

#### 3.1.2. NETest Levels and Follow-Up Assessment

The median NETest score was 40% (IQR: 33.3–46.7%); 71%, 9%, and 18% of patients were classified as belonging to the low-, medium-, and high-risk category, respectively (Table 2). As per the NETest score, 67 (66%) patients were classified as having stable disease while 35 (34%) patients were deemed to have progressive disease. Among the patients with NETest-determined disease progression at the time of NETest collection, 20 (57.1%) had simultaneous radiological disease progression. No significant difference in NETest levels was observed between patients that displayed radiologically stable disease or progression at the time of NETest collection (*p* = 0.058).

The NETest value could predict radiological disease progression at the time of collection with a sensitivity of 42.9% and a specificity of 91.5% using an optimal cut-off value of 76.7%. It could predict disease progression during a follow-up time of 18 months with a sensitivity of 40.9% and a specificity of 86.7% using an optimal cut-off value of 56.7%. The assignment of a patient to the NETest high-risk category predicted progressive disease with a sensitivity of 40.0% and specificity of 86.8%. The AUC to differentiate stable from progressive disease at the time of NETest collection was 0.61 (95% CI: 0.33–0.88) and during follow-up was 0.62 (95% CI: 0.46–0.77) (Figure 1).

#### 3.1.3. Prognostic Relevance of NETest

The UVA and MVA demonstrated that the NETest value (HR: 1.032, 95% CI: 1.003–1.062, *p* = 0.033), the high-risk NETest category (HR: 10.5, 95% CI: 1.35–81.7, *p* = 0.025), the presence of lung metastases (HR: 5.21, 95% CI: 1.12–24.3, *p* = 0.035), CgA levels > 10 × ULN (HR: 17.4, 95% CI: 3.21–93.7, *p* = 0.001), and TGR (HR: 2.469, 95% CI: 1.186–5.139, *p* = 0.016) were statistically significantly and independently associated with shorter PFS (Table 3). The related Kaplan–Meier curves are shown in Figure 2A.

The UVA and MVA demonstrated that the NETest value (HR: 1.035, 95% CI: 1.005–1.066, *p* = 0.024), the high-risk NETest category (HR: 15.2, 95% CI: 1.52–151, *p* = 0.020), the presence of lung metastases (HR: 13.7, 95% CI: 2.08–90.6, *p* = 0.007), and CgA levels > 10 × ULN (HR: 12.5, 95% CI: 1.33–117, *p* = 0.027) were statistically significantly and independently associated with shorter OS (Table 4). The related Kaplan–Meier curves are shown in Figure 2B.

### 3.2. Group 2

#### 3.2.1. Patient and Tumour Characteristics

Sixteen patients with a negative ^68^Ga-DOTATATE PET after surgical treatment for Si-NEN were analysed (Table 1). The mean age was 59.3 ± 12.3 years, and the female-to-male ratio was 0.78 (7/9). All patients were diagnosed with well-differentiated neuroendocrine tumours. The WHO grade for Si-NENs was grade 1 in 11 patients (69%) and grade 2 in 5 patients (31%). According to the TNM staging system [25], one patient had localized disease limited to the subserosa, fifteen patients had lymph node metastases, and none had distant metastases. Full TNM staging was not available in one patient.

#### 3.2.2. NETest Levels and Follow-Up Assessment

The median NETest score was 26.7% (IQR: 26.7–40%); 11 (69%), 1 (7%), and 2 (14%) patients were classified as belonging to the low-, medium- and high-risk categories, respectively (Table 2). Two (12.5%) patients had normal NETest levels. As per the NETest, 14 (87.5%) patients were classified as having disease post-operatively, and 2 (12.5%) patients were classified as disease free.

Among the 14 (87.5%) patients that were deemed to have disease post-operatively as per the NETest, 5 displayed radiological disease recurrence; the median time to recurrence since surgery was 30 months. Among these five patients, four were identified with ^68^Ga-DOTATATE PET and one with conventional imaging. Three were classified as belonging to the low-risk NETest category and the remaining two in the medium- and high-risk categories each.

Of the 11 (69%) patients classified as belonging to the low-risk category, all except 2 remained free of disease during the follow-up time. The patient that was classified as belonging to the medium-NETest-risk category displayed disease recurrence on follow-up imaging. The two patients with high NETest scores (86.7% and 100%) had no evidence of disease recurrence on the ^68^Ga-DOTATATE PET performed at one year follow-up while one of them displayed disease recurrence on conventional cross-sectional imaging. The AUC for predicting disease recurrence at one-year follow-up after surgery was 0.909 (95% CI: 0.75–0.100) (Figure 3).

#### 3.2.3. Prognostic Relevance of NETest

The UVA demonstrated that the grade, the TNM stage, the CgA levels, NETest value, and the high-risk NETest category were not significantly associated with the recurrence-free survival (Table 5).

## 4. Discussion

A critical unresolved issue in the management of patients with NENs is that neither imaging evaluation nor the currently available circulating biomarkers can identify early post-operative recurrence or disease progression. Over the years, several blood biomarkers, general and specific, have been developed to assist physicians in the management of patients with NENs. However, the currently available conventional circulating biomarkers, such as CgA, have poor sensitivity, specificity, and predictive ability [26]. The majority of studies have shown that CgA displays moderate diagnostic accuracy while it has limited utility as a marker of morphological disease progression [21,26]. The United States National Cancer Institute summit, held in 2007 on NENs, documented that all biomarkers currently used in clinical practice are monoanalytes and do not accurately reflect the biological behaviour of NENs. Hence, the development of novel diagnostic, prognostic, and predictive biomarkers for NENs is a critical area of need [2].

Recently, new techniques have been developed making use of circulating multi-analyte biomarkers that reflect disease status and therapeutic efficacy with promising results. The NETest, a multi-analyte biomarker designed specifically for NENs, is the most extensively studied and has been observed to display remarkable sensitivity and specificity as a diagnostic and follow-up biomarker in gastro-entero-pancreatic NENs (GEP-NENs) [13]. It is based on the simultaneous measurement of 51 neuroendocrine specific marker genes in peripheral blood and through a series of mathematical algorithms, it provides a score of disease activity that classifies patients into clinical categories of low, medium and high risk [12]. In this study, the value of the NETest as a biomarker in patients with Si-NENS was assessed in two different groups of patients diagnosed with Si-NENs.

Multiple studies have shown the NETest to have superior diagnostic accuracy over CgA with a higher sensitivity (98%) and specificity (97%), while also being highly reproducible [20]. The diagnostic accuracy of NETest in patients with Si-NENs is 93% while recent studies have demonstrated the utility of NETest in the evaluation of complete surgical resection [14,19,20]. Modlin et al. assessed the value of NETest in 153 operated NEN patients and showed that a NETest score >20 predicted radiological disease recurrence with 94% accuracy [20]. Another prospective study of 13 patients with Si-NENs who underwent surgical resection of the primary tumour and/or mesenteric mass demonstrated that the NETest may accurately identify post-operative residual or progressive disease in this small cohort [15]. Furthermore, its attractive health economics have been demonstrated when post-surgical NETest follow-up stratification is adopted, showing cost reductions of 42% [27]. In our study, we analysed the data of 16 patients that were post-operatively disease-free based on negative Ga-DOTATATE PET results. Among the five patients that displayed NETest-determined disease recurrence, patients were deemed to have post-operative residual disease, and three were deemed to have progressive disease. Furthermore, in accordance with previous studies, ROC analysis showed that the NETest is an excellent tool for predicting post-operative disease recurrence in operated patients with Si-NENs. Of note, among patients that were classified per the NETest as having residual or progressive disease, a significant proportion did not display radiological recurrence during the one-year follow-up. We believe that these patients may have microscopic residual disease of an indolent nature and an extended follow-up duration is warranted to capture the potential radiological disease recurrence.

A recent meta-analysis demonstrated that the NETest was 84.5–85.5% accurate in differentiating stable disease from progressive disease [13]. In addition, Pavel et al. showed that the NETest was able to identify disease progression one year before imaging studies [21]. We studied 102 patients with metastatic well-differentiated Si-NENs with a median follow-up time of 18 months. ROC analysis demonstrated that the NETest had moderate accuracy in differentiating stable from progressive disease in this cohort of patients. However, this could be attributed to the relatively short follow-up time. A recent prospective multi-centre study assessing the value of NETest over a 5-year follow-up period in 1684 patients with NENs, demonstrated a diagnostic accuracy of 91% that significantly outperformed CgA while it accurately stratified RECIST-assessed disease status [20].

In previous studies, elevated NETest scores were associated with poor PFS [21,28]. In our study, NETest was found to be a prognostic factor for PFS and OS in patients with metastatic Si-NENs. Of note, only extremely elevated CgA levels (>10 × ULN) were identified as an independent prognostic factor for PFS and OS in this cohort.

TGR was significantly associated with PFS as per RECIST criteria. This supports its validity as a radiological prognostic biomarker for patients with Si-NENs. TGR could be a valuable resource particularly in tumours exhibiting rapid radiological enlargement. As TGR is calculated as a continuous output whereas RECIST criteria is binary, TGR can be useful as an additional tool for assessment of disease status and predictor of PFS.

There are several limitations to our study. The sample size is small, particularly in the post-operative disease-free group of patients. Furthermore, the findings should be interpreted with caution due to the relatively short follow-up time. As disease recurrence is commonly observed within the first five post-operative years, the percentage of patients that recur may be underestimated during a one-year follow-up time [28]. Taking also into account the relatively indolent course of Si-NENs, even in the presence of metastases, certainly extended follow-up is required to detect disease recurrence or progression in these neoplasms.

## 5. Conclusions

In conclusion, our study provides real-life data of NETest results in a prospective cohort of patients with Si-NENs and demonstrates that the multianalyte blood NETest scores were associated with PFS and OS in patients with metastatic Si-NENs. In addition, the NETest may facilitate the early prediction of disease recurrence in the post-operative setting. TGR, CgA > 10 × ULN, and the presence of lung metastases also have a role in predicting PFS or OS. Studies with extended follow-up are warranted to robustly assess the utility of the NETest in the early identification of disease recurrence or progression in patients with Si-NENs.

## Figures and Tables

**Figure 1 cancers-16-02506-f001:**
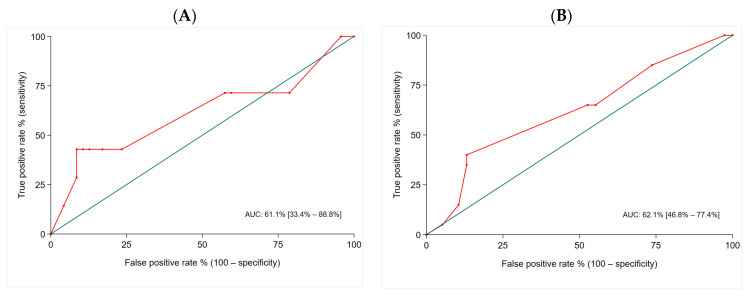
AUC for differentiating progressive from stable disease (**A**) at the time of NETest collection and (**B**) during follow-up in patients with metastatic Si-NENs.

**Figure 2 cancers-16-02506-f002:**
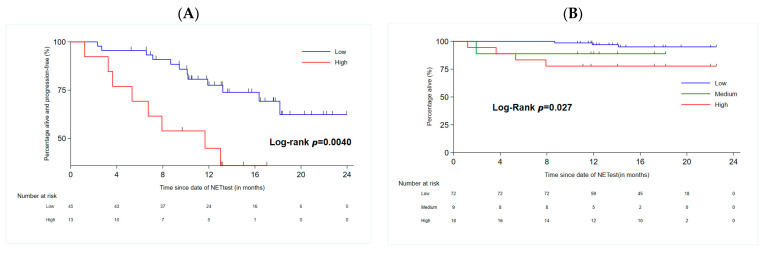
Kaplan–Meier curves of (**A**) progression-free survival and (**B**) overall survival, measured from time of NETest collection in patients with metastatic Si-NENs.

**Figure 3 cancers-16-02506-f003:**
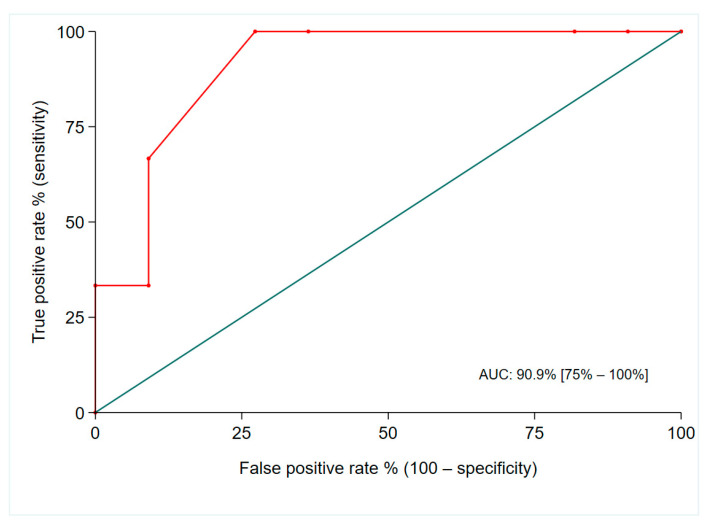
AUC for predicting post-operative disease recurrence in patients with operated Si-NENs.

**Table 1 cancers-16-02506-t001:** Patients’ baseline characteristics at the time of NETest collection (For continuous variables, if normally distributed, data displayed as mean ± SD, otherwise displayed as median and interquartile range). N/A = Not Applicable.

Characteristics	Group 1	Group 2
	Number (*n* = 102)	Number (*n* = 16)
Age	65.3 ± 10.2	59.3 ± 12.3
Gender		
Female	50 (49%)	7 (44%)
Male	52 (51%)	9 (56%)
Grade		
G1	59 (58%)	11 (69%)
G2	36 (35%)	5 (31%)
G3	0 (0%)	0 (0%)
Not available	7 (7%)	0 (0%)
Metastatic burden		N/A
Liver metastases	83 (81%)	
Lung metastases	13 (13%)	
Bone metastases	37 (36%)	
Other metastases	75 (74%)	
Treatments previously received		N/A
SSTA	90 (88%)	
PRRT	30 (29%)	
Everolimus/sunitinib	0 (0%)	
Liver embolization	7 (7%)	
Liver resection	9 (9%)	
Other surgery	16 (16%)	
Serum CgA		
<5 × ULN	55 (54%)	10 (62%)
5–10 × ULN	10 (10%)	1 (6%)
>10 × ULN	22 (22%)	2 (13%)
Not available	15 (15%)	3 (19%)
Urinary 5-HIAA		
<5 × ULN	11 (11%)	1 (6%)
5–10 × ULN	5 (5%)	0 (0%)
>10 × ULN	4 (4%)	0 (0%)
Not available	82 (80%)	15 (94%)
TNM stage	N/A	
T1N1Mx		2 (13%)
T2N1Mx		4 (25%)
T3N1Mx		4 (25%)
T4N1Mx		3 (19%)
T3N1M1		1 (6%)
T1N0Mx		1 (6%)
Unknown		1 (6%)

**Table 2 cancers-16-02506-t002:** NETest results and imaging evaluation at the time of NETest collection and during follow-up. (NETest is expressed as median and interquartile range).

NETest and Imaging Assessment	Group 1	Group 2
	Number (*n* = 102)	Number (*n* = 16)
NETest value (%)	40% (33.3–46.7%)	26.7% (26.7–40%)
NETest category		
Normal	3 (3%)	2 (13%)
Low	72 (71%)	11 (69%)
Medium	9 (9%)	1 (6%)
High	18 (18%)	2 (13%)
Progression at time of NETtest (CT/MRI)		N/A
Stable disease	47 (46%)	
Progression	7 (7%)	
Not available	48 (47%)	
Progression during follow-up (CT/MRI)		
Stable disease	47 (46%)	10 (63%)
Progression (group 1)/Recurrence (group 2)	15 (15%)	2 (13%)
Not available	40 (39%)	4 (25%)
Progression during follow-up (Ga-DOTATATE)		
Stable disease	17 (16.7%)	12 (75%)
Progression (group 1)/Recurrence (group 2)	14 913.7%)	4 (25%)
Not available	71 (69.6%)	0 (0%)

**Table 3 cancers-16-02506-t003:** Cox regression analysis of predictive factors for PFS since NETest collection in patients with metastatic Si-NETs.

Variables	Univariate		Multivariate	
	HR (95% CI)	*p* Value	HR (95% CI)	*p* Value
Age in decades	1.14 [0.76–1.71]	0.529		
Gender	Violates proportional hazards	Violates proportional hazards		
Male				
Female				
NETest value in %	1.018 [1.003–1.034]	0.020	1.032 [1.003–1.062]	0.033
NETest category				
Low	1		1	
High	3.57 [1.42–8.97]	0.007	10.5 [1.35–81.7]	0.025
Grade				
G1	1			
G2	2.41 [1.00–5.81]	0.051		
Liver metastasis *^,^**				
Absent	1			
Present	1.96 [0.40–125]	0.182		
Lung metastasis				
Absent	1		1	
Present	2.73 [1.06–7.02]	0.038	5.21 [1.12–24.3]	0.035
Bone metastasis				
Absent	1			
Present	1.38 [0.59–3.23]	0.462		
Serum CgA × ULN	1		1	
5–10 × ULN	1.96 [0.49–7.85]	0.343	0.32 [0.04–2.32]	0.260
>10 × ULN	4.45 [1.56–12.7]	0.005	17.4 [3.21–93.7]	0.001
Urinary 5HIAA *				
<5 × ULN	1			
5–10 × ULN	6.93 [0.19–255]	0.293		
>10 × ULN	10.6 [0.36–314]	0.171		
Progression at NETest				
Stable disease	1			
Progressive disease	2.25 [0.72–7.00]	0.161		
Tumour growth rate at NETest (%/month)	1.29 [1.10–1.50]	0.001	2.469 [1.186–5.139]	0.016
Liver burden				
No liver mets	1			
<25%	1.14 [0.30–4.32]	0.846		
25%–50%	2.36 [0.53–10.6]	0.262		
>50%	3.23 [0.80–13.0]	0.100		

* due to zero-event groups leading to non-convergence of cox regression, Firth’s penalised maximum likelihood method used. ** due to small number of patients without liver mets, inappropriate for multivariate analysis.

**Table 4 cancers-16-02506-t004:** Cox regression analysis of predictive factors for OS since NETest collection in patients with metastatic Si-NETs.

Variables	Univariate		Multivariate	
	HR (95% CI)	*p* Value	HR (95% CI)	*p* Value
Age in decades	1.45 [0.71–2.97]	0.305		
Gender				
Male	1			
Female	3.12 [0.63–15.5]	0.164		
NETest value in %	1.027 [1.003–1.051]	0.026	1.035 [1.005–1.066]	0.024
NETest category				
Low	1		1	
Medium	3.25 [0.34–31.5]	0.309	11.3 [0.63–203]	*p* = 0.099
High	6.15 [1.37–27.5]	0.018	15.2 [1.52–151]	*p* = 0.020
Grade				
G1	1			
G2	1.25 [0.28–5.58]	0.772		
Liver metastasis *				
Absent	1			
Present	4.18 [0.20–86.1]	0.355		
Liver burden				
No liver mets	1			
<25%	0.63 [0.04–10.2]	0.748		
25%–50%	6.02 [0.54–66.6]	0.143		
>50%	5.75 [0.60–55.6]	0.130		
Lung metastasis				
Absent	1		1	
Present	7.94 [1.98–31.8]	0.003	13.7 [2.08–90.6]	*p* = 0.007
Bone metastasis				
Absent	1			
Present	1.86 [0.47–7.45]	0.379		
Serum CgA *				
<5 × ULN	1		1	
5–10 × ULN	1.85 [0.05–66.3]	0.736	2.49 [0.05–117]	*p* = 0.641
>10 × ULN	10.0 [1.33–75.5]	0.025	12.5 [1.33–117]	*p* = 0.027
Progression at NETest				
Stable disease	1			
Progressive disease	1.80 [0.20–16.1]	0.600		

* due to zero-event groups leading to non-convergence of cox regression, Firth’s penalised maximum likelihood method used. 5HIAA not analysed as only 1 observation in entire cohort with valid 5HIAA measurement.

**Table 5 cancers-16-02506-t005:** Univariate Cox analysis of predictive factors for disease recurrence one year post NETest collection.

Variables (Progression Rate in Brackets)	HR (95% CI)	*p* Value
Age in decades	0.74 [0.30–1.81]	0.503
Gender		
Male (2/9)	1	
Female (3/7)	2.63 [0.30–23.0]	0.383
NETest category *		
Low (3/11)	1	
High (1/2)	2.67 [0.12–57.6]	0.532
Grade		
G1 (3/11)	1	
G2 (2/5)	1.78 [0.19–16.5]	0.613
Serum CgA		
Normal (3/7)	1	
Raised (1/6)	0.27 [0.02–3.65]	0.322
TNM stage **		
T1-2N1Mx (0/6)	1	
T3-4N1Mx (4/7)	16.7 [0.68–409]	0.084

* NETest value analysis was performed separately using logistic regression, OR 1.030 [0.982–1.081] *p* = 0.222. ** Firth’s penalized log likelihood used, as no progression in T1-2N1Mx (0/6) compared to T3-4N1Mx (4/7).

## Data Availability

The data presented in this study are available on request from the corresponding author due to sensitive patient data restricting its permission for open access.

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
