# Peer review of "The Clinical Utility of the NETest in Patients with Small Intestinal Neuroendocrine Neoplasms (Si-NENs): A “Real-Life” Study"

_cancers, 2024, doi:10.3390/cancers16142506_

Round 1
Reviewer 1 Report
Comments and Suggestions for Authors
Our fellow researchers propose a paper in which they reported the results of research on a single blood biomarker to adequately predict the behavior of neuroendocrine tumors. Starting from the concept that chromogranin A is not always reliable. We agree that these diseases often present themselves when they are advanced, given that they do not cause any symptoms and can be a casual finding during tests conducted for another pathology and only rarely cause skin flushes, diarrhea or hypertensive peaks, given that the hormones they produce often they are not complete or regularly configured so they are ineffective since they do not adhere to their respective cellular receptors. The therapy is surgical and medical. The second uses somatostatin or analogues which can cause complications (PMID: 38051513 please mention), Everolimus which has the positive effect of inhibiting calcineurin and therefore in case of renal failure and/or transplanted patients has a double positive use (DOI : 10.1016/j.trre.2015.09.001 if you want to cite it). Another surgical option is radionuclides always active on somatostatin receptors (DOI: 10.1016/j.ando.2019.04.005 if you want to cite it). We agree on the follow-up. We suggest, with the aim of studying the patient with imaging, the possibility of carrying out PET-MRI with 18F-6-fluoro-L-dihydroxyphenylalanine (18FDOPA, n=30), 18F-fluoro- 2-deoxy-D -glucose (18FDG, n=21), or 68Ga-(DOTA(0)-Phe(1)-Tyr(3))-octreotide (68Ga-DOTATOC, n=20) with T1-Dixon sequence simultaneous and diffusion-weighted imaging (DWI). Very interesting, however, is the NETest, on blood, which our colleagues propose to us, which allows us to monitor the progress of the disease over time with the pre-peri post distinctions with a detailed description of the conduct of the exam, the scoring and therefore reproducibility of the same. The study uses two distinct groups, one with simple disease, the second with metastatic disease. Harmonious discussion and absolutely irreproachable in its considerations, weak and strong points. we fully agree with the conclusions. the iconography supports the theses, materials and results very well. Bibliography needs to be implemented, English is good
Author Response
Our fellow researchers propose a paper in which they reported the results of research on a single blood biomarker to adequately predict the behavior of neuroendocrine tumors. Starting from the concept that chromogranin A is not always reliable. We agree that these diseases often present themselves when they are advanced, given that they do not cause any symptoms and can be a casual finding during tests conducted for another pathology and only rarely cause skin flushes, diarrhea or hypertensive peaks, given that the hormones they produce often they are not complete or regularly configured so they are ineffective since they do not adhere to their respective cellular receptors. The therapy is surgical and medical. The second uses somatostatin or analogues which can cause complications (PMID: 38051513 please mention), Everolimus which has the positive effect of inhibiting calcineurin and therefore in case of renal failure and/or transplanted patients has a double positive use (DOI : 10.1016/j.trre.2015.09.001 if you want to cite it). Another surgical option is radionuclides always active on somatostatin receptors (DOI: 10.1016/j.ando.2019.04.005 if you want to cite it). We agree on the follow-up. We suggest, with the aim of studying the patient with imaging, the possibility of carrying out PET-MRI with 18F-6-fluoro-L-dihydroxyphenylalanine (18FDOPA, n=30), 18F-fluoro- 2-deoxy-D -glucose (18FDG, n=21), or 68Ga-(DOTA(0)-Phe(1)-Tyr(3))-octreotide (68Ga-DOTATOC, n=20) with T1-Dixon sequence simultaneous and diffusion-weighted imaging (DWI). Very interesting, however, is the NETest, on blood, which our colleagues propose to us, which allows us to monitor the progress of the disease over time with the pre-peri post distinctions with a detailed description of the conduct of the exam, the scoring and therefore reproducibility of the same. The study uses two distinct groups, one with simple disease, the second with metastatic disease. Harmonious discussion and absolutely irreproachable in its considerations, weak and strong points. we fully agree with the conclusions. the iconography supports the theses, materials and results very well. Bibliography needs to be implemented, English is good
Thank you for your positive comments.
As suggested, I have included the reference below in the manuscript with the following added sentence in the introduction:
“SSAs are largely well tolerated; some adverse effects include injection site pain, steatorrhoea and more significantly cholelithiasis”
Calomino N, Poto GE, Carbone L, et al. Neuroendocrine tumors' patients treated with somatostatin analogue could complicate with emergency cholecystectomy. Ann Ital Chir. 2023;94:518-522.
Reviewer 2 Report
Comments and Suggestions for Authors
In their study, Julian Gertner and coworkers aimed to evaluate NETest in real-life as a biomarker for early identification of tumor progression and assessment of risk for post-operative recurrence in patients with small intestinal neoplasms.
The manuscript is clear, well-written, and informative.
I have only some minor points. As for the linguistic suggestions, since English native-speaking authors are included, please consider my “suggestions” simply as possible alternatives in the text.
SIMPLE SUMMARY, ABSTRACT
In the Simple Summary, the authors state that “NETest scores were associated with progression-free survival (PFS) and overall survival (OS) in metastatic Si-NENs.” It is not clear which part of the remaining text (except the presence of lung metastasis) applies to metastatic Si-NENs, non-metastatic Si-NENs, or both.
Similar considerations apply to the abstract section.
Additionally, does the term “early recurrence” refer only to metastatic spread, or does it include local recurrence as well?
Regarding “Metastatic Si-NENs,” since group 2 is defined as “post-operatively 68Ga disease-free,” should we assume that patients in group 1 all have 68Ga-positive metastasis?
Definition of group 2: “post-operatively 68Ga disease-free Si-NENs,” and “with no post-operatively macroscopic disease.” Would the authors consider providing a single, consistent definition?
Simple summary:
Suggestion:
“early recurrence” instead of “recurrence early.”
Abstract:
Suggestion:
“according to” instead of “as per.”
INTRODUCTION
Suggestions:
“include” instead of “includes.”
“difficulty in detecting” instead of “difficulty detecting.”
“most commonly used blood biomarker, but limitations…” instead of “most commonly used blood biomarker but limitations…”
“reflect disease activity, therapeutic efficacy, and improve early detection” instead of “reflect disease activity and therapeutic efficacy, to improve the early detection.”
AIM OF THE STUDY
The aim of the study is defined as:
Validity of NETest, in real life, as a biomarker for:
1.early identification of tumor progression
2. assessment of risk for post-operative recurrence
Was the term “validity” used as a generic term, or in its specific statistical meaning (the extent to which a test accurately measures what it is intended to measure)?
Could the authors confirm that all the results (i.e. OS, PFS) are related to the reported aims, and that the aim of the study is consistently reported in the manuscript?
RESULTS
73% of patients in group 1 and 31% of patients in group 2 had an unknown primary. Were they classified as Si-NENs based on anatomopathological features?
Suggestions:
“All patients were diagnosed as” instead of “All patients were diagnosed with.”
(Same suggestion on page 9)
“…low, medium, and high-risk categories respectively (Table 2);” a comma before “respectively” might be considered.
DISCUSSION
Suggestion: “Radiological” or “radiologic”?
Comments on the Quality of English Language
Minor editing of English language required
Author Response
In their study, Julian Gertner and coworkers aimed to evaluate NETest in real-life as a biomarker for early identification of tumor progression and assessment of risk for post-operative recurrence in patients with small intestinal neoplasms.
The manuscript is clear, well-written, and informative.
I have only some minor points. As for the linguistic suggestions, since English native-speaking authors are included, please consider my “suggestions” simply as possible alternatives in the text.
SIMPLE SUMMARY, ABSTRACT
Comment 1
In the Simple Summary, the authors state that “NETest scores were associated with progression-free survival (PFS) and overall survival (OS) in metastatic Si-NENs.” It is not clear which part of the remaining text (except the presence of lung metastasis) applies to metastatic Si-NENs, non-metastatic Si-NENs, or both.
Response 1
Thank you for your helpful comments.
We have made changes to the Simple Summary denoting more clearly if the text is in reference to the subgroup with metastatic Si-NENs (Group 1) or the subgroup who are post-operatively disease-free as per 68Ga-DOTATATE PET assessment (Group 2).
Comment 2
Similar considerations apply to the abstract section.
Response 2
We have reviewed the abstract which states:
“We studied two patient groups: Group 1: metastatic Si-NENs (N=102); Group 2: post-operatively disease-free as per 68Ga-DOTATATE PET (N=16).”
Comment 3
Additionally, does the term “early recurrence” refer only to metastatic spread, or does it include local recurrence as well?
Response 3
The term “recurrence” refers to either metastatic or local recurrence and this has been defined more clearly in the manuscript.
Comment 4
Regarding “Metastatic Si-NENs,” since group 2 is defined as “post-operatively 68Ga disease-free,” should we assume that patients in group 1 all have 68Ga-positive metastasis?
Response 4
It is not unreasonable to assume that all patients in Group 1 had 68Ga-DOTATATE PET positive metastases but this data was unfortunately not collected.
Comment 5
Definition of group 2: “post-operatively 68Ga disease-free Si-NENs,” and “with no post-operatively macroscopic disease.” Would the authors consider providing a single, consistent definition?
Response 5
Thank you for highlighting the typographical discrepancy for the Simple Summary compared with the Abstract. This was a conscious choice given that the Simple Summary should be written in layman’s terms so we avoided using “68Ga-DOTATATE PET”. Whereas, for the abstract we used the more precise definition.
Simple summary:
Suggestion 1:
“early recurrence” instead of “recurrence early.”
We have changed to “NETest monitoring after surgery may have potential in helping to predict early local or metastatic disease recurrence”
Abstract:
Suggestion 2:
“according to” instead of “as per.”
We have updated the manuscript to “according to” as per your suggestion.
INTRODUCTION
Suggestion 3:
“include” instead of “includes.”
We have changed the manuscript to “Systemic treatment options for non-resectable metastatic disease include long-acting somatostatin analogues (SSAs), everolimus and peptide receptor radionuclide therapy (PRRT)”
Suggestion 4:
“difficulty in detecting” instead of “difficulty detecting.”
We have changed the manuscript to “Functional imaging with 68Ga-DOTATATE positron emission tomography (68Ga-DOTATATE PET) also has limitations such as reduced spatial resolution (5 mm) and difficulty in detecting marginal change of tumour size”
Suggestion 5:
“most commonly used blood biomarker, but limitations…” instead of “most commonly used blood biomarker but limitations…” –
We have changed to “Chromogranin A (CgA) has been the most important and the most commonly used blood biomarker, but limitations in assay reproducibility, sensitivity and specificity have been extensively documented”
Suggestion 6:
“reflect disease activity, therapeutic efficacy, and improve early detection” instead of “reflect disease activity and therapeutic efficacy, to improve the early detection.”
We have changed to “There is a need for novel multianalyte biomarkers that can accurately reflect disease activity, therapeutic efficacy and improve early detection of recurrent and progressive disease.”
AIM OF THE STUDY
The aim of the study is defined as:
Validity of NETest, in real life, as a biomarker for:
1.early identification of tumor progression
2. assessment of risk for post-operative recurrence
Comment 5:
Was the term “validity” used as a generic term, or in its specific statistical meaning (the extent to which a test accurately measures what it is intended to measure)?
Response 5:
The term “validity” was used primarily as a generic term. Although, by virtue of the fact statistical analyses were run on the NETest for early identification of tumour progression and assessment of risk for post-operative recurrence, its accuracy was also being directly assessed.
Comment 6
Could the authors confirm that all the results (i.e. OS, PFS) are related to the reported aims, and that the aim of the study is consistently reported in the manuscript?
Response 6
As per the manuscript, we state:
“The aim of this prospective study was to evaluate the validity of the NETest, in real-life, as a biomarker for the early identification of tumour progression and assessment of risk for post-operative recurrence in patients with Si-NENs.”
We conclude that:
“our study provides real-life data of NETest results in a prospective cohort of patients with Si-NENs and demonstrates that the multianalyte blood NETest scores were associated with PFS and OS in patients with metastatic Si-NENs.”
However, we acknowledge that larger prospective studies with extended follow-up are required to confirm these findings.
RESULTS
Comment 7
73% of patients in group 1 and 31% of patients in group 2 had an unknown primary. Were they classified as Si-NENs based on anatomopathological features?
Response 7
Patients discussed at the Royal Free Hospital MDM will have multimodal assessments to contribute to a diagnosis of Si-NEN. A combination of clinical history, biological behaviour, imaging, histopathology and crucially immunohistochemistry, helps in classifying patients as Si-NENs when the primary is unknown.
Suggestions 7:
“All patients were diagnosed as” instead of “All patients were diagnosed with.”
(Same suggestion on page 9)
Thank you for this comment. “All patients were diagnosed with” is grammatically correct.
Suggestion 8:
“…low, medium, and high-risk categories respectively (Table 2);” a comma before “respectively” might be considered.
Thank you, we have changed this in 2 places for consistency:
“NETest levels and follow-up assessment
The median NETest score was 26.7% (IQR: 26.7-40%); 11 (69%), 1 (7%) and 2 (14%) patients were classified as low, medium and high-risk categories, respectively (Table 2).”
“NETest levels and follow-up assessment
The median NETest score was 40% (IQR: 33.3-46.7%); 71%, 9% and 18% of patients were classified as low, medium and high-risk categories, respectively (Table 2).”
DISCUSSION
Suggestion 9:
“Radiological” or “radiologic”?
Thank you for raising this point. Radiological and radiologic are equally correct in the medical literature and we have not changed the manuscript on this occasion.